# Probabilistic Connection Importance Inference and Lossless Compression of Deep Neural Networks

**Xin Xing**
Harvard University

**Long Sha**
Brandeis University

**Pengyu Hong**
Brandeis University

**Zuofeng Shang**
New Jersey Institute of Technology

**Jun S. Liu**
Harvard University

## Abstract

Deep neural networks (DNNs) can be huge in size, requiring a considerable amount of energy and computational resources to operate, which limits their applications in numerous scenarios. It is thus of interest to compress DNNs while maintaining their performance levels. We here propose a probabilistic importance inference approach for pruning DNNs. Specifically, we test the significance of the relevance of a connection in a DNN to the DNN's outputs using a nonparemetric scoring test and keep only those significant ones. Experimental results show that the proposed approach achieves better lossless compression rates than existing techniques.

## 1 Introduction

Deep neural networks (DNNs) have many successful applications ranging from computer vision, natural language processing to computational biology. However, large DNNs usually require significant memory and storage overhead and sometimes a large network bandwidth, which hinges their usages on mobile devices. Running large-size neural networks also consumes a considerable amount of energy, making their deployments on battery-constrained devices difficult. Furthermore, the over-parameterization issue in DNN architectures can impair its performances. Recent works (Han et al. (2015); Ullrich et al. (2017); Louizos et al. (2017) and references therein) show ways to reduce the network complexity by using proper regularization or network pruning to significantly reduce the number of parameters. One way to convert a dense DNN into a sparse one is by applying $L_0/L_1$ regularization on the model parameters. The $L_1$ penalty is computationally efficient, but it also introduces more bias on the large weights and may lead to significant degradation in model performances (Han et al., 2015). In contrast, $L_0$ regularization shows better performance, but incurs much higher computational complexity due to its combinatorial nature. Pruning, as shown in Han et al. (2015) and Tartaglione et al. (2018), is another promising strategy to sparsify DNNs by only keeping network connections more relevant to the final output. The importance of a network connection (i.e., the connection between two network nodes) can be approximated by the magnitudes or gradients of its weights. However, such an approximation may not be accurate since it does not consider the highly nonlinear relationships between network connections and the final output induced by the multi-layer convolutions of DNNs.

Some available network compression methods improve the computational performance with moderate to high losses in accuracy, which can be highly undesirable in many critical missions (such as autonomous driving). In order to achieve lossless compression, we need to correctly decipher the relationships between the network connections and the final output. This is a challenging task because the structural nature of DNNs makes the dependence between a network connection and the network output highly nonlinear. In this paper, we propose a probabilistic connection importance inference (PCII) method for testing whether a connection in a DNN is relevant to the DNN output. Specifically, we introduce a new technique called probabilistic tensor product decomposition to decompose the association of two connected network nodes into two components: one related to the DNN output and the other independent of the DNN output. If the strength of the first component is high, we keep the

network connection. Otherwise, we delete it. The inference is carried out by a nonparametric score test, which is based on modeling the log-transformed joint density of the connected nodes and the final output in a tensor product reproducing kernel Hilbert space (RKHS). We further derive the asymptotic distribution of the proposed test statistics, thus avoiding the computationally intensive resampling step encountered in most nonparametric inference settings. We implemented the method and tested it on image classification tasks, in which our approach achieved the highest lossless compression rates.

Section 2 reviews relevant literature; Section 3 introduces the PCII method and algorithm; Section 4 establishes theoretical properties for using the method to infer dependence between a network connection and the DNN output; and Section 5 shows the experimental results and concludes with a short discussion.

## 2  RELATED WORKS

Zhu & Gupta (2017) found that a DNN can be greatly sparsified with minimal loss in accuracy. One strategy for sparsifying DNNs is to shrink small weights to zero. Along this line of thinking, Louizos et al. (2017) introduced a smoothed version of $L_0$ regularization aiming to be both computationally feasible and beneficial to generalizability. There are also some regularization methods based on Bayesian formulations. Ullrich et al. (2017) proposed to add a Gaussian mixture prior to the weights. The sparsity is achieved by concentrating weights to cluster centers. Molchanov et al. (2017) proposed variational dropout, which learns individual dropout rate based on the empirical Bayes principle. Also, as shown in Han et al. (2015), the performance of pruning and retraining is related to choice the correct regularization, such as $L_1$ or $L_2$ regularization.

PCII offers a means to prune a DNN by keeping network connections that are the most relevant to the DNN output. This idea goes back to the optimal brain damage work by LeCun et al. (1990). It shows that among the many parameters in the network, many are redundant and do not contribute significantly to the output. Later, Hassibi et al. (1993) proposed the optimal brain surgeon method, which leverages a second-order Taylor expansion to select parameters for deletion. Most recently, Han et al. (2015) proposed a three-step pruning method. Tartaglione et al. (2018) proposed to prune a DNN based on the sensitivity score. There are also several approaches targeting at sparsifying convolution layers. For example, Anwar et al. (2017) proposed to prune feature maps and kernels in convolution layers.

Comparing with existing pruning methods based on the magnitude or gradient of weights, our approach directly models the relationship between a network connection and the network output in a nonparametric way. In addition, our inference is based on a statistical hypothesis testing, which outputs $p$-values to quantify the dependence strength of each network connection to the DNN output. The use of $p$-values allows network connections to be treated in a unified way and alleviates the need of *ad hoc* weights normalization required in some existing approaches.

## 3  PROBABILISTIC CONNECTION IMPORTANCE INFERENCE

In this section, we establish a general procedure for building the probabilistic connection structure, in which the connections are inferred by the nonparametric score test in tensor product RKHS. As shown in our experiments (see Section 5), our technique not only sparsifies the network but improves its generalization ability.

### 3.1  PCII ON FULLY CONNECTED LAYERS

Without loss of generality, we let a feed-forward acyclic DNN (Figure 1) be composed of $T$ layers with $X_t$ being the input of the $t$-th network layer, where $t = 0, 1, \ldots, T$. Let $t = 0$ and $t = T$ indicate the input and output layers, respectively, and let $0 < t < T$ index the hidden layers. The collection of all the nodes is denoted as $\mathcal{G}$ and the collection of all network connections is denoted as $\mathcal{E}$. We use a pair of nodes to denote a connection in $\mathcal{E}$. For example, $(X_{t,1}, X_{t+1,1})$ denotes an edge from the $1^{st}$ node in the $t$-th layer to the $1^{st}$ node in the $(t + 1)$-th layer. For simplicity, we use $Y$ to denote the output layer, who takes on categorical values for a classification task, and takes on continuous values for regression.

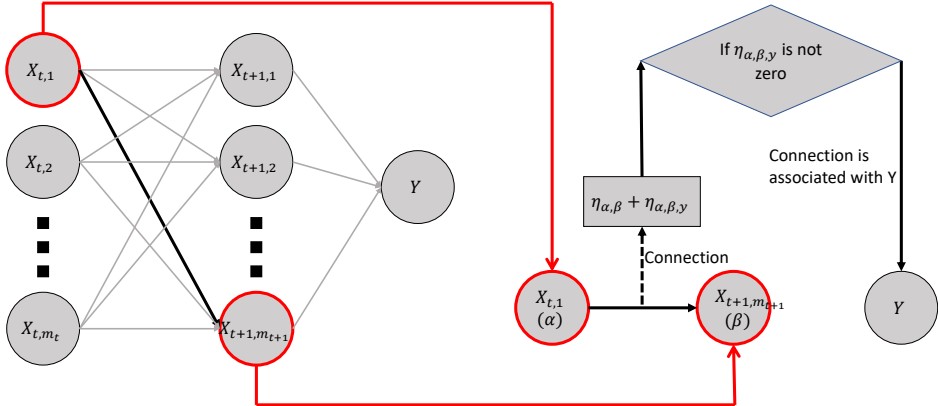

Figure 1: Flowchart for probabilistic connection inference. We have two nodes $\alpha$ and $\beta$ from a fully connected neural network. The connection of $\alpha$ and $\beta$ is expressed as $\eta_{\alpha,\beta} + \eta_{\alpha,\beta,y}$. The importance of the connection is inferred by testing whether the three way interaction $\eta_{\alpha,\beta,y}$ is zero or not.

The output of the $(t+1)$-th layer can be described as

$$X_{t+1,j} = g_{t+1}(\sum_{i=1}^{m_t} w_{t,t+1}^{i,j} X_{t,i}), \text{ for } j = 1, \ldots, m_{t+1},$$

where $m_t$ denotes the number of nodes in the $t$-th layer, and $g_{t+1}$ is the activation function. It should be noted that the magnitude of weights is not necessarily the best indication of the impact of the corresponding network connection. A more preferable way is to directly model the relationship between a network connection and the final output of a DNN. For simplicity, we use $(\alpha, \beta)$ to denote an arbitrary network connection. Due to the high non-linearity of a DNN, we use the nonparametric function to model the relationship among $\alpha$, $\beta$ and $Y$.

Let us denote the joint density of $\alpha$, $\beta$ and $Y$ as $f(\alpha, \beta, y)$ and the log-transferred density as $\eta(\alpha, \beta, y)$. We assume that $\eta(\alpha, \beta, y)$ can be deposed as

$$\eta(\alpha, \beta, y) = \eta_\alpha + \eta_\beta + \eta_y + \eta_{\alpha,y} + \eta_{\beta,y} + \underbrace{\eta_{\alpha,\beta} + \eta_{\alpha,\beta,y}}_{\text{interaction between } \alpha \text{ and } \beta}, \tag{1}$$

where $\eta_\alpha$, $\eta_\beta$, and $\eta_y$ are marginal effects, $\eta_{\alpha,y}$, $\eta_{\beta,y}$, and $\eta_{\alpha,\beta}$ are the two-way interaction terms, $\eta_{\alpha,\beta,y}$ is the three-way interaction term. Here, we interpret the connection as the interaction of $\alpha$ and $\beta$, i.e. $\eta_{\alpha,\beta} + \eta_{\alpha,\beta,y}$. Specifically, $\eta_{\alpha,\beta}$ is the interaction effect of $\alpha$ and $\beta$ without the impact of $y$, and $\eta_{\alpha,\beta,y}$ characterizes the interaction of $\alpha$, $\beta$ impacted by $y$. Our aim is to measure the significance of the connection by how much it is related to $Y$. To model this problem mathematically, we measure the association between the connection and $Y$ by the significance of the three-way interaction $\eta_{\alpha,\beta,y}$. Therefore, inferring whether the connection is related to the final output $Y$ or not is equivalent to testing whether $\eta_{\alpha,\beta,y} = 0$ or not. As shown in Figure 1, the connection is important for the network model if and only if the three-way interaction $\eta_{\alpha,\beta,y} \neq 0$. We propose a score test to qualify the significance of this term. The detailed construction of the statistical test is explained in Section 4.2.

## 3.2 PCII on convolutional layers

For different activation functions and type of layers, we have modifications to adopt the specific structure. Here, we generalize the proposed PCII test to handle convolutional layers, which are critical in many modern neural network architectures such as VGG16 Simonyan & Zisserman (2014). As demonstrated in Li et al. (2016), sparsifying the filter has little effect on reducing the computation cost. Nevertheless, reducing the volume of filters can greatly increase computing efficiency. For example, the volume of a filter is $3 \times 3 \times 4$. There are four $3 \times 3$ slices. PCII can be generalized to infer the importance of the slices, which can be treated as the connection between one channel in the current layer and one channel in the next layer.

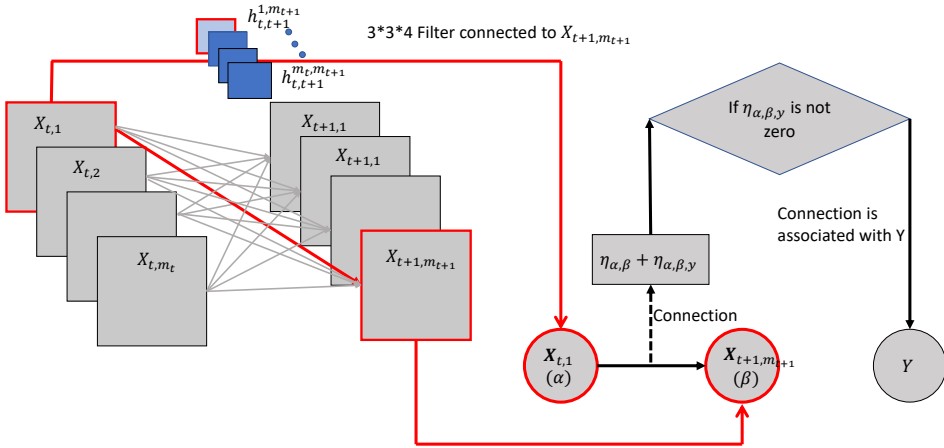

Figure 2: Flowchart for probabilistic connection inference for convolutional layers. Without loss generality, we use same notation of $\alpha$ and $\beta$ to denote two channels from a convolutional neural network. The connection between $\boldsymbol{\alpha}$ and $\boldsymbol{\beta}$ is corresponding to the convolutional operator $h_{t,t+1}^{1,m_{t+1}}$ (shown as one slice with the red border). The importance of this connection is inferred by testing whether the three way interaction $\eta_{\boldsymbol{\alpha},\boldsymbol{\beta},y}$ is zero or not.

Convolutional filters can be applied to transform channels in the $t$-th layer to channels in the $(t+1)$-th layer. We denote the $j$-th channel in the $t$-th layer as $\boldsymbol{X}_{t,j}$ for $j = 1, \ldots, m_t$, where $m_t$ is the number of channels in $t$-th layer and $t = 0, \ldots, T$. Each filter connects all channels in the $t$-th layer to one channel in the $t+1$-th layer. Let $h_{t,t+1}^{i,j}$ denote a convolution operator that connects two channels $\boldsymbol{X}_{t,i}$ and $\boldsymbol{X}_{t+1,j}$. Then, the filter connected to $\boldsymbol{X}_{t+1,j}$ is $\{h_{t,t+1}^{i,j}\}_{i=1}^{m_t}$ where we denote each $h_{t,t+1}^{i,j}$ as a slice of the filter. For example, when we choose a $3 \times 3 \times 4$ filter and set stride as one, this operator is the filter convolved (slided) across the width and height of the input $\{\boldsymbol{X}_{t,i}\}_{i=1}^{m_t}$ and offset with the bias. For one slice in the $t+1$-th layer, we can write its connection with the slices in the previous layer as

$$\boldsymbol{X}_{t+1,j} = g_{t+1}\left(\sum_{i=1}^{m_t} h_{t,t+1}^{i,j}(\boldsymbol{X}_{t,i}) + b_{t,t+1}^j\right)$$

where $g_{t+1}$ is an activation function and $b_{t,t+1}^j$ is the bias for the $j$-th filter. As illustrated in Figure 2, the red arrow from $X_{t,1}$ to $X_{t+1,m_{t+1}}$ represents $h_{t,t+1}^{1,m_{t+1}}$, which is one slice of the filter $\{h_{t,t+1}^{i,m_{t+1}}\}_{i=1}^{m_t}$ connecting the channels in the $t$-th layer to channel $X_{t+1,m_{t+1}}$ in the $t+1$-th layer. For simplicity, we denote $\alpha$ as one channel in the current layer and $\beta$ as one channel in the next layer. Since the relationship among the triplet $(\alpha, \beta, Y)$ is highly nonlinear, we model its log-transformed joint density as a nonparametric function $\eta(\alpha, \beta, y)$. Similar to the fully connected layers, we assume that $\eta(\alpha, \beta, y)$ has a decomposition as in (1). As shown in Figure 2, the connection between $\alpha$ and $\beta$ is decomposed to two parts: one unrelated to the output $Y$, $\eta_{\alpha,\beta}$, and another related to the output $Y$, $\eta_{\alpha,\beta,y}$. We aim to select connections that have greater contributions to $Y$, which is mathematically translated into the task of assessing the significance of the three-way interaction term $\eta_{\alpha,\beta,y}$ against the null hypothesis $\eta_{\alpha,\beta,y} = 0$.

### 3.3 ALGORITHM

In real applications, both fully connected layers and convolutional layers are used in a complex neural network architecture. We integrate the PCII procedure described in the previous two subsections to simultaneously infer the connections in both fully connected and convolutional layers, as summarized in Algorithm 1. We use $p_{ij}^{(t)}$ to denote the $p$-value for testing the $i$-th node in $t$-th layer and $j$-th node in $(t+1)$-th layer for the fully connected layers. For convolutional layers, $p_{ij}^{(t)}$ denotes the p-value for inferring the importance of the filter connecting the $i$-th slice in the $t$-th layer and the $j$-th slice in

the $t + 1$-th layer. The calculation of the p-values are given in Section 4 based on our proposed Score test.

---

**Algorithm 1** PCII: Probabilistic Connection Importance Inference for Lossless Neural Network Compression

---

**Input:** A training dataset, a DNN architecture, and the desired model compression rate
**Step 1:** Use the training dataset to train a DNN of the given architecture.
**Step 2:** (a). Importance inference of connections in fully connected layers: infer the three-way dependence effect of a network connection by testing the hypothesis $H_0 : \eta_{\alpha,\beta,y} = 0$ v.s $H_1 : \eta_{\alpha,\beta,y} \neq 0$, and calculate the test statistics (or $p$-values) for all network connections as $\boldsymbol{p}^f = \{p_{i,j}^{t,t+1}, i = 1, \ldots, m_t, j = 1, \ldots, m_{t+1} \mid t$ and $t + 1$ layers are fully connected}.
(b). Importance inference of connections in convolutional layers: infer the three-way dependence effect of a network connection by testing the hypothesis $H_0 : \eta_{\alpha,\beta,y} = 0$ v.s $H_1 : \eta_{\alpha,\beta,y} \neq 0$, and calculate the test statistics (or $p$-values) for all network connections as $\boldsymbol{p}^c = \{p_{i,j}^{t,t+1}, i = 1, \ldots, m_t, j = 1, \ldots, m_{t+1} \mid t$ and $t + 1$ layers are convolutionally connected}.
**Step 3:** Rank all network connections by their test statistics (or $p$-values), and select a threshold $\rho^f$ for deleting network connections in fully connected layers and $\rho^c$ in convolutional layers to achieve the desired model compression rate (Strategies for choosing $\rho^f$ and $\rho^c$ are given in Section 5).
**Step 4:** Set the same initial value for non-zero weights and filters. Retrain the sparsified DNN.

---

## 4 SCORE TEST AND THEORETICAL PROPERTIES

### 4.1 BACKGROUND ON TENSOR PRODUCT RKHS

Consider two random variables $\alpha$ and $\beta$ for fully connected layers or two random vectors $\alpha$ and $\beta$ for convolutional layers. Let $Y$ be a random variable as the final output. The domains for $\alpha, \beta$ and $Y$ are $\mathcal{A}, \mathcal{B}$, and $\mathcal{Y}$, respectively. Here, we assume that the log-transformed joint density function $\eta$ belongs to a tensor product RKHS $\mathcal{H} = \mathcal{H}^{\langle\alpha\rangle} \otimes \mathcal{H}^{\langle\beta\rangle} \otimes \mathcal{H}^{\langle Y\rangle}$ where $\otimes$ denotes the tensor product of two linear space.

For marginal RKHS, $\mathcal{H}^{\langle l\rangle}$ with an inner product $\langle\cdot,\cdot\rangle_{\mathcal{H}^{\langle l\rangle}}$ for $l = \alpha, \beta, Y$, there exists a symmetric and square integrable function $K_l$ such that

$$\langle f, K_l(x,\cdot)\rangle_{\mathcal{H}^{\langle l\rangle}} = f(x), \text{ for all } f \in \mathcal{H}^{\langle l\rangle} \tag{2}$$

where $K_l$ is called the reproducing kernel of $\mathcal{H}^{\langle l\rangle}$ for $l = \alpha, \beta, Y$. By Mercer's theorem, any continuous kernel has the following decomposition

$$K(x,y) = \sum_{\nu=0}^{\infty} \mu_\nu \phi_\nu(x)\phi_\nu(y) \tag{3}$$

where the $\mu_\nu$'s are non-negative descending eigenvalues and the $\phi_\nu$'s are eigen-functions. For the discrete domain $\{1, \ldots, a\}$, we define the kernel as $K(x,y) = \mathbb{1}\{x = y\}$ for $x, y \in \{1, \ldots, a\}$. For a continuous domain, there are different choice of kernels such as Gaussian kernels and Sobolev kernels. Note that the eigenvalues for different kernels on continuous domain have different decay rate. For example, the eigenvalues of the Gaussian kernel have the exponential decay rate, i.e., there exists some $c > 0$ such that $\mu_\nu \asymp \exp(-c\nu)$ (Sollich & Williams, 2004); and the eigenvalues of the $m$-th Sobolev kernels have the polynomial decay rate, i.e., $\mu_\nu \asymp \nu^{-2m}$ (Gu, 2013).

### 4.2 PROBABILISTIC DECOMPOSITION OF TENSOR PRODUCT RKHS

Next we propose the probabilistic tensor sum decomposition for each marginal RKHS, $\mathcal{H}^{\langle l\rangle}$, for $l = \alpha, \beta, Y$. We first use Euclidean space as a simple example to illustrate the basic idea of tensor sum decomposition. Recall that the tensor sum decomposition is often called ANOVA decomposition in linear model. For example, for the $d$-dimensional Euclidean space, we denote $f$ as a vector and let $f(x)$ be the $x$-th entry of the vector for $x = 1, \ldots, d$. We denote $\mathcal{A}$ as an average operator defined as $\mathcal{A}f(x) = \langle\frac{1}{d}\mathbf{1}, f\rangle$. The tensor sum decomposition of the Euclidean space $\mathbb{R}^d$ is

$$\mathbb{R}^d = \mathbb{R}_0^d \oplus \mathbb{R}_1^d := span\{\frac{1}{d}\mathbf{1}\} \oplus \{f \in \mathbb{R}^d \mid \sum_{i=1}^{d} f(d) = 0\} \tag{4}$$

where the first space is called the grand mean and the second space is called the main effect. Then, we construct the kernel for $\mathbb{R}_0^d$ and $\mathbb{R}_1^d$ in the following lemma.

**Lemma 1.** *For a RKHS space $\mathcal{H}$, there corresponds a unique reproducing kernel $K$, which is non-negative definite. Based on the tensor sum decomposition $\mathcal{H} = \mathcal{H}_0 \oplus \mathcal{H}_1$, where $\mathcal{H}_0 = \{1/d\mathbf{1}\}$ and $\mathcal{H}_1 = \{g \in \mathcal{H} : E_X(g(X)) = 0\}$, we have the kernel for $\mathcal{H}_0$ as*

$$k_0(x,y) = 1/d \tag{5}$$

*and kernel for $\mathcal{H}_1$ as*

$$k_1(x,y) = \mathbb{1}_{\{x=y\}} - 1/d$$

*where $\mathbb{1}$ denotes the indicator function.*

However, in RKHS with infinite dimension, the grand mean is not a single vector. Here, we set the average operator $\mathcal{A}$ as $\mathcal{A} := f \to E_x f(x) = E_x \langle k_x, f \rangle_{\mathcal{H}} = \langle E k_x, f \rangle_{\mathcal{H}}$ where $k$ is the kernel function in $\mathcal{H}$ and the first equality is due to the reproducing property. $E_x k_x$ plays the same role as $\frac{1}{d}\mathbf{1}$ in Euclidean space. Then, we have the tensor sum decomposition in a functional space defined as

$$\mathcal{H} = \mathcal{H}_0 \oplus \mathcal{H}_1 := span\{E_x k_x\} \oplus \{f \in \mathcal{H} : \mathcal{A}f = 0\}. \tag{6}$$

Following the same fashion, we call $\mathcal{H}_0$ as the grand mean space and $\mathcal{H}_1$ as the main effect space. Notice that $E_x k_x$ is also known as the kernel mean embedding which is well established in the statistics literature Berlinet & Thomas-Agnan (2011). Then we introduce the following lemma to construct the kernel function for $\mathcal{H}_0$ and $\mathcal{H}_1$.

**Lemma 2.** *For RKHS space $\mathcal{H}$, there corresponds an unique reproducing kernel $K$, which is non-negative definite. Based on the tensor sum decomposition $\mathcal{H} = \mathcal{H}_0 \oplus \mathcal{H}_1$ where $\mathcal{H}_0 = \{E_x k_x\}$ and $\mathcal{H}_1 = \{g \in \mathcal{H} : E_x(g(x)) = 0\}$, we have the kernel for $\mathcal{H}_0$ as*

$$k_0(x,y) = E_x[k(x,y)] + E_y[k(x,y)] - E_{x,y}k(x,y) \tag{7}$$

*and the kernel for $\mathcal{H}_1$ as*

$$\begin{aligned} k_1(x,y) &= \langle k_x - E k_x, k_y - E k_y \rangle_{\mathcal{H}} \\ &= k(x,y) - E_x[k(x,y)] - E_y[k(x,y)] + E_{x,y}k(x,y). \end{aligned}$$

In neural networks, $\mathcal{A}$, $\mathcal{B}$ are in a continuous domain. The final output $\mathcal{Y}$ can be either in continuous domain or discrete domain, which depends on the tasks. Here, we construct the tensor sum decomposition for discrete domain and continuous domain based on Lemma 1 and Lemma 2 respectively. Specifically, we have spaces $\mathcal{H}^{\langle \alpha \rangle}$, $\mathcal{H}^{\langle \beta \rangle}$ and $\mathcal{H}^{\langle Y \rangle}$ decomposed as tensor sums of subspaces $\mathcal{H}^{\langle \alpha \rangle} = \mathcal{H}_0^{\langle \alpha \rangle} \oplus \mathcal{H}_1^{\langle \alpha \rangle}$, $\mathcal{H}^{\langle \beta \rangle} = \mathcal{H}_0^{\langle \beta \rangle} \oplus \mathcal{H}_1^{\langle \beta \rangle}$, $\mathcal{H}^{\langle Y \rangle} = \mathcal{H}_0^{\langle Y \rangle} \oplus \mathcal{H}_1^{\langle Y \rangle}$. Following Gu (2013), we apply the distributive law and have the decomposition of $\mathcal{H}$ as

$$\begin{aligned} \mathcal{H} =& (\mathcal{H}_0^{\langle \alpha \rangle} \oplus \mathcal{H}_1^{\langle \alpha \rangle}) \otimes (\mathcal{H}_0^{\langle \beta \rangle} \oplus \mathcal{H}_1^{\langle \beta \rangle}) \otimes (\mathcal{H}_0^{\langle Y \rangle} \oplus \mathcal{H}_1^{\langle Y \rangle}) \\ \equiv& \mathcal{H}_{000} \oplus \mathcal{H}_{100} \oplus \mathcal{H}_{010} \oplus \mathcal{H}_{001} \oplus \mathcal{H}_{110} \oplus \mathcal{H}_{101} \oplus \mathcal{H}_{011} \oplus \mathcal{H}_{111}. \end{aligned} \tag{8}$$

where $\mathcal{H}_{ijk} = \mathcal{H}_i^{\langle \alpha \rangle} \oplus \mathcal{H}_j^{\langle \beta \rangle} \oplus \mathcal{H}_k^{\langle Y \rangle}$.

**Lemma 3.** *Suppose $K^{\langle 1 \rangle}$ is the reproducing kernel of $\mathcal{H}^{\langle 1 \rangle}$ on $X_1$, and $K^{\langle 2 \rangle}$ is the reproducing kernel of $\mathcal{H}^{\langle 2 \rangle}$ on $X_2$. Then the reproducing kernels of $\mathcal{H}^{\langle 1 \rangle} \otimes \mathcal{H}^{\langle 2 \rangle}$ on $X = X_1 \times X_2$ is $K(\boldsymbol{x}, \boldsymbol{y}) = K^{\langle 1 \rangle}(x^{\langle 1 \rangle}, y^{\langle 1 \rangle})K^{\langle 2 \rangle}(x^{\langle 2 \rangle}, y^{\langle 2 \rangle})$ with $\boldsymbol{x} = (x^{\langle 1 \rangle}, x^{\langle 2 \rangle})$ and $\boldsymbol{y} = (y^{\langle 1 \rangle}, y^{\langle 2 \rangle})$.*

Lemmas 3 can be easily proved based on Theorems 2.6 in Gu (2013). Lemma 3 implies that the reproducing kernels of the tensor product space is the product of the reproducing kernels. Based on these three lemmas, we can construct kernel for each subspace defined in (8).

## 4.3 SCORE TEST

Based on (8), the log-transformed density function $\eta \in \mathcal{H}$ can be correspondingly decomposed as (1). Thus, $\eta_{\alpha,\beta,Y} = 0$ if and only if $\eta^* \in \mathcal{H}_0 := \mathcal{H}_{000} \oplus \mathcal{H}_{100} \oplus \mathcal{H}_{010} \oplus \mathcal{H}_{001} \oplus \mathcal{H}_{110} \oplus \mathcal{H}_{101} \oplus \mathcal{H}_{011}$ where $\eta^*$ is the underlying truth. Hence, we focus on the following hypothesis testing problem:

$$H_0 : \eta^* \in \mathcal{H}_0 \text{ vs. } H_1 : \eta^* \in \mathcal{H} \backslash \mathcal{H}_0, \tag{9}$$

where $\mathcal{H}\backslash\mathcal{H}_0$ denotes set difference of $\mathcal{H}$ and $\mathcal{H}_0$. We next propose a likelihood-ratio based procedure to test (9). Suppose that $\boldsymbol{t} = (\alpha, \beta, y)$ and $\boldsymbol{t}_i = (\alpha_i, \beta_i, y_i)$, $i = 1, \ldots, n$, are *iid* observations generated from $\mathcal{T} = (\mathcal{A}, \mathcal{B}, \mathcal{Y})$. Let $LR_n$ be the likelihood ratio functional defined as

$$LR_n(\eta) = \ell_n(\eta) - \ell_n(P_{\mathcal{H}_0}\eta) = -\frac{1}{n}\sum_{i=1}^{n}\{\eta(\boldsymbol{t}_i) - P_{\mathcal{H}_0}\eta(\boldsymbol{t}_i)\}, \ \eta \in \mathcal{H}, \tag{10}$$

where $P_{\mathcal{H}_0}$ is a projection operator from $\mathcal{H}$ to $\mathcal{H}_0$. Using the reproducing property, we rewrite (10) as

$$LR_n(\eta) = -\frac{1}{n}\sum_{i=1}^{n}\{\langle K_{\boldsymbol{t}_i}, \eta\rangle_{\mathcal{H}} - \langle K_{\boldsymbol{t}_i}^0, \eta\rangle_{\mathcal{H}}\} \tag{11}$$

where $K$ is the kernel for $\mathcal{H}$ and $K^0$ is the kernel for $\mathcal{H}_0$.

Then we calculate the Fréchet derivative of the likelihood ratio functional as

$$DLR_n(\eta)\Delta\eta = \left\langle \frac{1}{n}\sum_{i=1}^{n}(K_{\boldsymbol{t}_i} - K_{\boldsymbol{t}_i}^0), \Delta\eta \right\rangle_{\mathcal{H}} = \left\langle \frac{1}{n}K_{\boldsymbol{t}_i}^1, \Delta\eta \right\rangle_{\mathcal{H}} \tag{12}$$

where $K^1$ is the kernel for $\mathcal{H}_{111}$. We define our test statistics as the squared norm of the score function of the likelihood ratio functional as

$$S_n^2 = \|\frac{1}{n}\sum_{i=1}^{n}K_{\boldsymbol{t}_i}^1\|_{\mathcal{H}}^2 \tag{13}$$

By the reproducing property, we can expand the left hand side of (13) as

$$S_n^2 = \frac{1}{n^2}\sum_{i=1}^{n}\sum_{j=1}^{n}K^1(\boldsymbol{t}_i, \boldsymbol{t}_j) \tag{14}$$

We observe an interesting phenomenon that $S_n^2$ has a similar expression with the MMD (Gretton et al., 2012) when $Y$ is a binary variable. When $Y \in \{0, 1\}$, the kernel on $\mathcal{H}^{\langle Y\rangle}$ is $K_1^{\langle Y\rangle}(y_i, y_j) = \mathbb{1}\{x = y\} - 1/2$. Assume that the kernel for $\mathcal{H}_1^{\langle\alpha\rangle} \otimes \mathcal{H}_1^{\langle\beta\rangle}$ is $K_{11}^{\langle\alpha,\beta\rangle}(\boldsymbol{x}, \boldsymbol{x})$ for $\boldsymbol{x} \in \mathcal{A} \times \mathcal{B}$. By Lemma 3, we have $K^1(\boldsymbol{t}_i, \boldsymbol{t}_j) = K_1^{\langle Y\rangle}(y_i, y_j)K_{11}^{\langle\alpha,\beta\rangle}(\boldsymbol{x}_i, \boldsymbol{x}_j)$. Then we have $8S_n^2$ as

$$8S_n^2 = \frac{4}{n^2}(\sum_{\{i,j \,|\, y_i=y_j=0\}} K_{11}^{\langle\alpha,\beta\rangle}(\boldsymbol{x}_i, \boldsymbol{x}_j) - 2\sum_{\{i,j \,|\, y_i\neq y_j\}} K_{11}^{\langle\alpha,\beta\rangle}(\boldsymbol{x}_i, \boldsymbol{x}_j) + \sum_{\{i,j \,|\, y_i=y_j=1\}} K_{11}^{\langle\alpha,\beta\rangle}(\boldsymbol{x}_i, \boldsymbol{x}_j)). \tag{15}$$

If we replace $K_{11}^{\langle\alpha,\beta\rangle}$ with $K^{\langle\alpha,\beta\rangle}$, i.e., the kernel on the $\mathcal{H}^{\langle\alpha\rangle} \otimes \mathcal{H}^{\langle\beta\rangle}$, the right hand side of (15) is the MMD measuring the distance between the joint distribution of $\alpha$ and $\beta$ in the group with $y = 0$ and the joint distribution of $\alpha$ and $\beta$ in the group with $y = 1$.

$$\text{MMD}^2[(\alpha, \beta), Y] = \frac{4}{n^2}(\sum_{\{i,j \,|\, y_i=y_j=0\}} K^{\langle\alpha,\beta\rangle}(\boldsymbol{x}_i, \boldsymbol{x}_j) - 2\sum_{\{i,j \,|\, y_i\neq y_j\}} K^{\langle\alpha,\beta\rangle}(\boldsymbol{x}_i, \boldsymbol{x}_j) + \sum_{\{i,j \,|\, y_i=y_j=1\}} K^{\langle\alpha,\beta\rangle}(\boldsymbol{x}_i, \boldsymbol{x}_j)) \tag{16}$$

Since we want to infer the importance of the connection between the $\alpha$ and $\beta$, we are only interested in comparing whether the interaction effect between $\alpha$ and $\beta$ changes or not in the two groups. The main effects of $\alpha$ and $\beta$ only contribute to the importance of the nodes or slices and are not relevant to the connection between $\alpha$ and $\beta$.

### 4.4 CALCULATION OF THE TEST STATISTICS

We introduce a matrix form of the squared norm of score function for manifesting the computation process. In (14), $S_n^2$ is determined by the kernel on th $\mathcal{H}_1^{\langle\alpha\rangle} \otimes \mathcal{H}_1^{\langle\beta\rangle} \otimes \mathcal{H}_1^{\langle Y\rangle}$. By Lemma 1 and Lemma 2, we replace the expectation by the sample average and get the kernel function for $\mathcal{H}_l^{\langle l\rangle}$ as

$$k_1^l(x, y) = k^l(x, y) - \frac{1}{n}\sum_{i=1}^{n}k^l(x_i, y) - \frac{1}{n}\sum_{i=1}^{n}k^l(x, y_i) + \frac{1}{n^2}\sum_{i=1}^{n}\sum_{j=1}^{n}k^l(x_i, y_j)$$

where $k^l$ is the kernel function for $\mathcal{H}^{\langle l \rangle}$ for $l = \alpha, \beta, Y$. Some popular choices of the kernel functions are Gaussian kernel, Laplace kernel and polynomial kernel. Let $K^l$ be the empirical kernel matrix. We can rewrite (14) as

$$S_n^2 = \frac{1}{n^2}[(HK^\alpha H) \circ (HK^\beta H) \circ (HK^Y H)]_{++} \tag{17}$$

where $H = I_n - \frac{1}{n}\mathbf{1}\mathbf{1}^T$, $I_n$ is a $n \times n$ identity matrix and $\mathbf{1}_n$ is a $n \times 1$ vector of ones, and $[A]_{++} = \sum_{i=1}^n \sum_{j=1}^n A_{ij}$. This test statistics is also related to three-way Lancaster's interaction measure (Lancaster, 2002).

Fortunately, the computation of (17) only involves the matrix multiplication, the computational procedure can be executed in parallel on GPU. In our experiment, the highly parallelized matrix operation tremendously alleviated the computing time: essentially reducing from quadratic to almost linear. In addition, we can reduce the sample size by sub-sampling $r << n$ data points. Sub-sampling the dataset is a popular technique to reduce computational burden and can efficiently approximate the full data likelihood in a regression setting (Ma et al., 2015; Drineas et al., 2011). However, for very large data set, the sub-sampling is not efficient. We consider a mini-batch algorithm to calculate the score in each batch and used the averaged score as the final test statistics. This is also related to the divide-and-conquer method which is widely used in kernel-based learning (Zhang et al., 2013; Shang & Cheng, 2017; Liu et al., 2018).

### 4.5 Asymptotic distribution

The asymptotic distribution of $S_n^2$ depends on the decay rate of the kernel of the product of RKHS. Suppose that $\{\mu_\nu^{\langle\alpha\rangle}, \phi_\nu^{\langle\alpha\rangle}\}_{\nu=1}^\infty$ is a series of eigenvalue and eigenfunction pairs for $\mathcal{H}_1^{\langle\alpha\rangle}$, $\{\mu_\nu^{\langle\beta\rangle}, \phi_\nu^{\langle\beta\rangle}\}_{\nu=1}^\infty$ is a sequence of basis for $\mathcal{H}_1^{\langle\beta\rangle}$. If $Y$ is continuous, we suppose that $\mathcal{H}^{\langle Y \rangle}$ has the eigensystem, $\{\mu_\nu^{\langle Y \rangle}, \phi_\nu^{\langle\alpha\rangle}\}_{\nu=1}^\infty$. If $Y$ is categorical, we suppose that $\mathcal{H}^{\langle Y \rangle}$ has the eigensystem, $\{\mu_\nu^{\langle Y \rangle}, \phi_\nu^{\langle\alpha\rangle}\}_{\nu=1}^{a-1}$. Then we have the eigenvalue eigenfuntion pair for the tensor product space $\mathcal{H}_1^{\langle\alpha\rangle} \otimes \mathcal{H}_1^{\langle\beta\rangle} \otimes \mathcal{H}_1^{\langle Y \rangle}$ as

$$\{\mu_{\nu_\alpha}^{\langle\alpha\rangle}\mu_{\nu_\beta}^{\langle\beta\rangle}\mu_{\nu_Y}^{\langle Y \rangle}, \phi_{\nu_\alpha}^{\langle\alpha\rangle}\phi_{\nu_\beta}^{\langle\beta\rangle}\phi_{\nu_Y}^{\langle Y \rangle}\}$$

where $\nu_\alpha = 1, \ldots, \infty$, $\nu_\beta = 1, \ldots, \infty$, $\nu_Y = 1, \ldots, \infty$ (Y is continuous), and $\nu_Y = 1, \ldots, a-1$ (Y is categorical with $a$ categories). For simplicity, we order the pairs in the decreasing order of $\mu_\rho$, $\rho = 1, \ldots, \infty$. The null hypothesis could be interpreted as factorization hypothesis, i.e., $(X, Y) \perp Z \vee (X, Z) \perp Y \vee (Y, Z) \perp X$ or X, Y, Z are mutually independent.

**Theorem 1.** *Suppose the kernel on $\mathcal{H}_{111}$ is square integrable. If $Y$ is continuous variable, then under $H_0$, we have*

$$nS_n^2 \xrightarrow{d} \sum_{\rho=1}^\infty \mu_\rho \epsilon_\rho^2 \tag{18}$$

*where $\epsilon_\rho$ are i.i.d. standard Gaussian random variables.*

The proof of this theorem is shown in the Supplementary Materials A.1. The asymptotic distribution of $nS_n^2$ only depends on the eigenvalues of the kernel. Theorem 1 is related to Wilks' phenomenon demonstrated in the classic nonparametric/semiparametric regression framework (Fan et al., 2001; Shang & Cheng, 2013), i.e., the asymptotic distribution is independent of the nuisance parameters. In practice, we fix the same kernel for the fully connected layers and the same kernel for the convolutional layers. Thus, it provides a unified importance measure for all connections in the same type of layer, which avoids the scaling problem faced by those pruning methods that use magnitudes of weights as an importance measure. In addition, we can use the value of the test statistics as an importance measure for pruning and bypass the effort of calculating the p-values since the order is the same according to either of these two measures.

## 5 Results

We conducted experiments to test out the PCII method in two supervised image classification tasks: MNIST and CIFAR10. We used TensorFlow to train DNNs. The back-end of PCII was implemented in Fortran and R language. Programming interfaces were implemented to connect the back-end calculations to TensorFlow. The experiments were run on a computer with one Nvidia Titan V GPU and 48 CPU cores.

PCII offers a convenient way to adjust compression rate by changing the $p$-value threshold $\rho$. For other compression methods in consideration, the compression rate is usually controlled by some hyper-parameters, which need to be tuned via an ad hoc trial and error strategy. Two types of comparisons were carried out. In the first type, we adjust the compression rate of a method while requiring its compressed DNN to achieve the test accuracy of the original uncompressed DNN. We term this the lossless compression rates (LCR). Since it is very time-consuming to obtain an exact test accuracy, we allow a 0.01% deviation from the desired test error rates. In the second type, we compared the minimum testing error (MTE) of the compressed DNNs produced by different model compression methods. MTE shows how a compression method can help increase the generalizability and robustness of a DNN. We only included the results of the tested methods that we could obtain their working codes to reproduce the results reported in their original papers. In addition, we did not include methods that are not able to achieve the LCR of the corresponding test dataset.

## 5.1 MNIST DATASET

We tried two network architectures for MNIST (60k training images and 10k test images): a multilayer perceptron (MLP) with two hidden layers of sizes 300 and 100, respectively, and a convolutional neural network LeNet-5 (LeCun et al. (1998)). We trained LeNet-300-100 for 100 epochs to achieve a test error of 1.69%. These approaches include two regularization based methods in Louizos et al. (2017), as well as several pruning based methods in Han et al. (2015), and Guo et al. (2016). The results are summarized in Table 1. PCII achieved the lowest MTE when the compression rate was 10x. Then, we further increased the compression rate until the error rate reached 1.70%. The results show that PCII not only compressed a medium-sized MLP better than existing methods but also was able to improve generalizability of a MLP via compression (i.e., the MTE is better than the LCR).

| Dataset: MNIST,  Network: LeNet-300-100 | | | |
|---|---|---|---|
| Criterion | Method | Error % | Compression Rate |
| | Original | 1.69% | 1x |
| LCR | PCII | 1.70% | **26x** |
| | Han et al. (2015) | 1.69% | 15.1x |
| | Louizos et al. (2017) | 1.70% | 3.3x |
| | Guo et al. (2016) | 1.70% | 15x |
| MTE | PCII | **1.58%** | 10x |
| | Han et al. (2015) | 1.59% | 12.2x |
| | Louizos et al. (2017) | 1.70% | 3.3x |
| | Guo et al. (2016) | 1.70% | 15x |

Table 1: Experimental results for LeNet-300-100 on MNIST dataset.

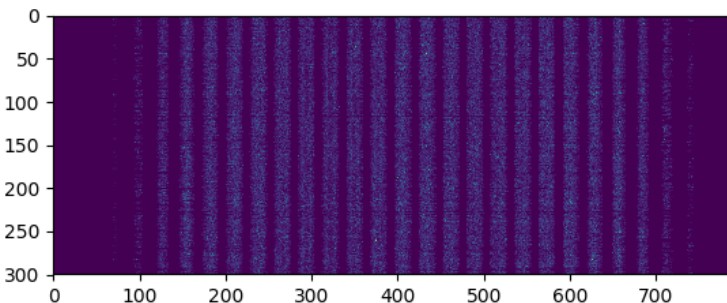

Figure 3: The heatmap showing the PLR test result of the $784 \times 300$ connections between the input layer and the first FC layer in Lenet-300-100. Each pixel represents a $p$-value of the corresponding pair. The brighter color representing a smaller $p$-value. The width and height of the heatmap correspond to the input dimension (784) and the size (300) of the first FC layer.

Interestingly, our inference results can also help in interpreting the fitted neural network. For example, through inferring the importance of the connections between input layer and the the first hidden layer, we can visualize the importance of the features in the input layer. Figure 3 plots the $p$-values of the

associations between the network connections in the first layer and the final output. The heatmap shows that a banded structure repeated 28 times, in which the central region tends to have smaller $p$-values. The left and right margins of the heatmap show that the connections on the first and last few channels are less relevant to the final output (i.e., have large $p$-values). This phenomenon is observed because a written digit is usually located in the central part of a image.

| | Dataset: MNIST, Network: LeNet-5 | | |
|---|---|---|---|
| Criterion | Method | Error % | Compression Rate |
| | Original | 0.70% | 1x |
| LCR | PCII | 0.69 % | **38x** |
| | Han et al. (2015) | 0.71% | 12.1x |
| | Louizos et al. (2017) | 0.69% | 1.4x |
| | Guo et al. (2016) | 0.70% | 32x |
| MTE | PCII | **0.65%** | 10.7x |
| | Han et al. (2015) | 0.70% | 6x |
| | Louizos et al. (2017) | 0.68% | 1.1x |
| | Guo et al. (2016) | 0.70% | 32x |

Table 2: Experimental results for LeNet-5-caffe on MNIST dataset.

The LeNet-5 model (`https://goo.gl/4yI3dL`) is a modified version of LeCun et al. (1998). It includes two convolutional layers with 20 and 50 filters, respectively, and a fully connected layer with 500 nodes. The results are summarized in Table 2. PCII achieved both the lowest MTE and highest LCR, again, for this model, demonstrating the broad applicability of the PCII strategy for various neural network architectures.

## 5.2 CIFAR10 DATASET

To demonstrate the applicability of PCII to complex DNNs, we applied it to VGG16 (Zagoruyko, 2015), which was adapted for the CIFAR-10 dataset (Krizhevsky & Hinton, 2009). The network consists of 13 convolutional and two fully-connected layers. The results are summarized in Table 3. The test error gradually decreases as the compression rate increases from 1x to 3x, and achieves the MTE at $6.01\%$. When the compression rate is further increased, the test error begins to increase. As the compression rate reaches 10x, the test error increases to $7.56\%$ that is comparable to the test error of the uncompressed VGG16. For this dataset, we only include the result for PCII due to the limited resources. In fact, we could not obtain the results for other methods in comparison in three days' computing time.

| | Dataset: CIFAR10, Network: VGG16 | | |
|---|---|---|---|
| Criterion | Method | Error | Compression Rate |
| | Original | 7.55% | 1x |
| LCR | PCII | 7.56% | 10x |
| MTE | PCII | 6.01% | 3x |

Table 3: Experimental results for VGG16 on CIFAR10 dataset.

## 6 DISCUSSION

We propose a statistically principled strategy to directly infer the importance of a connection in a neural network via PCII, a hypothesis testing framework. The proposed PCII test provides a p-value based measure on the importance of the connection through the connection's association with the final output. This type of direct quantification cannot be easily accomplished by the magnitude-based pruning method. Although the two examples are relatively small in size, they demonstrated the broad applicability of the PCII method and its improved power in network compression. Last but not least we note that the PCII testing method can be easily generalized to a broad class of connection types including the skip layer connections in RNN.

## 7 ACKNOWLEDGEMENT

XX and JL would like to acknowledge NSF and NIH for providing partial support or this work. PH and LS would like to acknowledge NSF (NSF OAC 1920147) for providing partial support of this work.

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

## A    SUPPLEMENTARY MATERIALS

### A.1    POOF OF THEOREM 1

Let $\{\mu_\rho, \phi_\rho\}_{\rho=1}^\infty$ be the eigenvalues and eigenvectors for the tensor product space $\mathcal{H}_1^{\langle\alpha\rangle} \otimes \mathcal{H}_1^{\langle\beta\rangle} \otimes \mathcal{H}_1^{\langle Y\rangle}$. By the dedomposition defined (8), we have

$$E_t[\phi_\rho(t)] = 0, \tag{19}$$

for $\rho = 1, \ldots, \infty$, i.e., the mean of the eigenfunction $\phi_\rho$ is zero. By simple calculation, we have

$$\sum_{\rho=1}^\infty \mu_\rho < \infty \tag{20}$$

for the exponential decayed kernels and polynomal decayed kernel with decay rate as $i^{-m}$ for $m > 1$. Thus, the commonly used kernels such as Gausssian kernel, Laplase Kernel and linear or qudratic Solblev kernel satisfy this requirement.

By Mercer's theorem, we have the decomposition as

$$
\begin{aligned}
nS_n^2 &= \frac{1}{n}\sum_{i=1}^n\sum_{j=1}^n K^1(\boldsymbol{t}_i, \boldsymbol{t}_j) \\
&= \sum_{i=1}^n\sum_{j=1}^n\sum_{\rho=1}^\infty \mu_\rho \phi_\rho(\boldsymbol{t}_i)\phi_\rho(\boldsymbol{t}_j) \\
&= \sum_{\rho=1}^\infty (\mu_\rho(\sum_{i=1}^n \phi(\boldsymbol{t}_i))^2) \\
&\xrightarrow{d} \sum_{\rho=1}^\infty \mu_\rho \epsilon_\rho
\end{aligned}
$$

where $\epsilon_\rho$ are i.i.d. standard normal random variables. The last row is proved by applying the Lindeberg–Lévy CLT to have $\sum_{i=1}^n \phi(\boldsymbol{t}_i) \xrightarrow{d} \epsilon_\rho$ since (1) and (2) holds. Then, by the Kolmogorov's inequality and $\sum_{\rho=1}^\infty \mu_\rho < \infty$, we have the last row holds.

