# OpenReview forum: "Probabilistic Connection Importance Inference and Lossless Compression of Deep Neural Networks"
_ICLR.cc/2020/Conference — Accept (Poster)_

### Official Review · AnonReviewer3 · 2019-10-23
**Official Blind Review #3**

**Rating:** 6

**Review:**

In this paper, the authors propose a new pruning technique that utilizes the statistical dependency between the corresponding nodes and outputs. The dependency is measured by a kernel based dependency measure which is closely related to MMD. The test statistics derived from the dependency measure have an asymptotic distribution which can be written as a weighted sum of chi-square random variables. The proposed method is numerically investigated using some datasets such as MNIST and CIFAR 10.

Pruning is one of important problems for practical deep learning operations. This paper gives an interesting idea for the pruning techniques. I think the idea is novel.

On the other hand, I also have the following concerns:
- Although applying the kernel type information measure is an interesting idea, its computational complexity would be large. It is not obvious that it works for large datasets such as ImageNet even if the sub-sampling technique is applied.
- The numerical experiments are conducted in small datasets. How it works in larger datasets such as ImageNet and (more importantly) how it is compared with SOTA pruning methods in more difficult datasets. Actually, performance comparison is not done for the CIRAR10-VGG16 setting.
- The asymptotic distribution can be seen as a corollary of existing researches for MMD and HSIC.

For these reasons, I was not completely convinced with the effectiveness of the proposed method.


Minor comment:
- The test statistics is more like HSIC. It would be nice if there were comments on the connection to HSIC.

===
Update: The computational cost for this method seems not so much demanding, and could be applied to large data-set. The  resultant performance also seems useful. I think more convincing comparisons are needed. However, its idea seems interesting and its practicality is ensured to some extent. Thus, I have raised  my score.

**Experience Assessment:**

I have published one or two papers in this area.

**Review Assessment: Checking Correctness Of Derivations And Theory:**

I assessed the sensibility of the derivations and theory.

**Review Assessment: Checking Correctness Of Experiments:**

I assessed the sensibility of the experiments.

**Review Assessment: Thoroughness In Paper Reading:**

I read the paper at least twice and used my best judgement in assessing the paper.

---

> ### Author Response · Authors · 2019-11-13
> **Reply to Review #3 - 1**
>
> We appreciate the reviewer for suggestions and questions. We address the concerns below and add a mini-batch implementation that is very efficient in calculating the test statistics for large datasets.
>
>
> 1.
> Reviewer:
> Although applying the kernel type information measure is an interesting idea, its computational complexity would be large. It is not obvious that it works for large datasets such as ImageNet even if the sub-sampling technique is applied.
>
> Response:
> We thank the reviewer for pointing out the computational concerns in larger datasets. Our approach can be implemented in a mini-batch fashion. Basically, we divide data into mini-batches, calculate the test statistics within each mini-batch, and then average the statistics overall mini-batches to produce the test statistics of the whole dataset. The mini-batch size can be smaller than the sub-sampling size and hence reduce the computational cost further. Theoretically, as shown in [1], the type-I error and power are guaranteed when the batch size is greater than the polynomial order of log(n) for the Gaussian kernel.   A similar technique was used in kernel-based deep learning methods, such as GMMN [2] and MMD-GAN [3]. In practice, the performance of the averaged test statistics based on the mini-batches is comparable with the oracle one calculated by using the full data, when the mini-batch size is carefully chosen (see [1] for details). Details of the implementation can be found in our revised manuscript Section 4.4. We report the running time of our method using sub-sampling and mini-batch on a workstation with a 48-core CPU@2.60GHz:
>
> =============================================================
>                                         Lenet-300-100     Lenet-5-Caffe     CIFAR10-VGG16
> sub-sampling (mins)            39.6                     26.8                    312.9
> Mini-batch  (mins)                29.6                     21.8                    103.9
> =============================================================
>
> The computational time of our mini-batch implementation improves dramatically over the sub-sampling implementation, especially for larger datasets such as CIFAR10. Due to the limitation of computational resources, we are not able to complete the experiments on ImageNet. So far the intermediate results look very promising and the final results will be included in our revision. We claim the computation cost is manageable for larger datasets and is of less concern for the following reasons:
> a. Users of model compression techniques (such as ours) only need to compress a model once, which can be conducted on large computer clusters to deal with the computational cost. Then the compressed model can be deployed to many devices with limited memory benefits.
> b.  Our sub-sampling or mini-batch implementations can significantly reduce computational cost.
>
> c. The statistical testing procedure can be implemented parallelly based on GPUs to further improve computing efficiency.
>
>
> [1] Liu et al. How Many Machines Can We Use in Parallel Computing for Kernel Ridge Regression?, arXiv,  2019.
> [2] Li et al. Generative moment matching networks. NeurlPS. 2015.
> [3] Li et al. "MMD GAN: Towards deeper understanding of moment matching network." NeurlPS. 2017.

---

> ### Author Response · Authors · 2019-11-14
> **Reply to Review #3 - 2**
>
>
> 2.
> Reviewer:
> The numerical experiments are conducted in small datasets. How it works in larger datasets such as ImageNet and (more importantly) how it is compared with SOTA pruning methods in more difficult datasets. Actually, performance comparison is not done for the CIRAR10-VGG16 setting.
>
> Response:
> We added the comparison results for the CIRAR10-VGG16 setting as follows:
>
> =========================================
>                                             ER% (*)               CR
> ———————————————————————
> LCR      PCII                         -                          10x
>              Li 17’ [1]                 -                          2.7x
>              Qianghui 18’ [2]   -                          5.8x
>              Babajide 18’ [3]    -                          4.5x
>              Wining Ticket [4]  -                          5x
> ———————————————————————
> MTE     PCII                         6.01 (+1.54)       3x
>              Li 17’ [1]                 6.60 (+0.15)       2.7x
>              Qianghui 18’ [2]   6.67 (-0.60)        5.8x
>              Babajide 18’ [3]     6.33 (-0.13)       4.5x
>              Wining Ticket [4]   6.10 (+0.30)      2x
>              Louizos17’ [5]        8.60 (-1.05)       14x
> =========================================
>
> * the result in the parenthesis is the relative accuracy change compared to its own baseline, larger is better.
>
> ** We reached out to the author of Louizos 2017 [5] about replicating the results. They told us their experiments usedGaussian process HP optimization and took a few weeks to finish. Our approach only took 312.9 mins in this setting.
>
> Again, due to the limited computational resources, we can obtain now, we are not able to complete the experiments on ImageNet. So far the intermediate results look very promising and the final results will be included in our revision.
>
>
> 3.
> Reviewer:
> The asymptotic distribution can be seen as a corollary of existing research for MMD and HSIC.
>
> Response:
> Our hypothesis testing procedure is novel in the sense that it is based on the probabilistic decomposition of a tensor RKHS. Following this line, our work can be extended to higher-order interactions. The test statistics are derived by the likelihood principle which is linked to the well-established work on minimax optimal nonparametric estimation. It leaves space to extend our work to regularized likelihood functionals by adding roughness penalty to the likelihood function which may improve the power and lead to the minimax optimal test. Section 4.3 reveals that MMD and HSIC are a special case of our likelihood ratio based score test for interactions, and hence the asymptotic distribution of our score test is consistent with the MMD and HSIC.
>
>
> 4.
> Reviewer:
> The test statistics are more like HSIC. It would be nice if there were comments on the connection to HSIC.
>
> Response:
> We thank the reviewer for pointing out the connection. MMD can be treated as a measure of dependence between a continuous random variable and a discrete random variable. HSIC essentially measures the dependence between two continuous variables.  Neither MMD nor HSIC covers the high-order dependence structures. Our proposed likelihood-based method models high-order dependence through functional decomposition.  Both MMD and HSIC can be deemed as special cases of our proposed framework.  The above discussion will be elaborated in the revised version.
>
>
> [1] Li, Hao, et al. Pruning filters for efficient convnets. arXiv preprint arXiv:1608.08710 (2016).
> [2] Huang, Qiangui, et al. Learning to prune filters in convolutional neural networks. 2018 IEEE Winter Conference on Applications of Computer Vision (WACV). IEEE, 2018.
> [3] Ayinde, Babajide O., and Jacek M. Zurada. Building efficient convnets using redundant feature pruning. arXiv preprint arXiv:1802.07653. 2018.
> [4] Liu, Zhuang, et al. Rethinking the value of network pruning. arXiv preprint arXiv:1810.05270 2018.
> [5] Louizos et al. Bayesian compression for deep Learning. NeurlPS. 2017.

---

### Official Review · AnonReviewer2 · 2019-10-25
**Official Blind Review #2**

**Rating:** 6

**Review:**

The paper proposes a new way of evaluating the importance of neural connections which can be used for better model compression. The approach uses a non-parametric statistical test to detect the three way interaction among the two nodes and final output. Some small scale experiments show that the approach achieves better compression rate given the same test error.

The approach seems interesting in the sense that, unlike existing techniques, it explicitly measures the three way interaction among the two nodes and the output. Also, it removes the average effects and only considers (non-linear) correlation after removing the mean. The explicit link to the final output and removing average effects allows the method to remove more weights without decreasing the loss by much.

However, the drawback of the approach is the significantly increased computation due to 1) quadratic complexity for the kernel methods; 2) unable to cache computation among different pairs of nodes (for example, gradient-based approach can compute importance for all node connections in one forward-backward pass). This limits the applicability of the approach to more interesting cases of larger models (for example, models that work on ImageNet) where model compression is of more urgent need. As a result, the significance and impact of the approach is also limited.

The experiment results are interesting in that it shows the proposed approach can achieve better compression rates given the same test error tolerance on a few small datasets. However, as mentioned above, these experiments are less convincing than more complicated models on larger datasets, such as ImageNet where model compression has greater impact. In addition, the test errors on smaller datasets are easier to achieve, sometimes tuning the optimization settings such as learning rates can result in significant improvement. Therefore, such results are more like preliminary.

The paper is general clear and well written. The experiment section should include more important information such as what kernel bandwidth is used for Gaussian kernel, which greatly affects performance. Also, the main text introducing the proposed statistical test is a bit verbose, and dense in unnecessary notations. I think it can be made more succinct by presenting a high level idea (removing mean, three way correlations) first and then the final results. Some intermediate results can be put into Appendix.

**Experience Assessment:**

I have published one or two papers in this area.

**Review Assessment: Checking Correctness Of Derivations And Theory:**

I assessed the sensibility of the derivations and theory.

**Review Assessment: Checking Correctness Of Experiments:**

I assessed the sensibility of the experiments.

**Review Assessment: Thoroughness In Paper Reading:**

I read the paper at least twice and used my best judgement in assessing the paper.

---

> ### Author Response · Authors · 2019-11-13
> **reply to Review #2 - 1**
>
> We appreciate the reviewer for suggestions and questions. We address the concerns below and updated the manuscript in which we added a mini-batch implementation to efficiently calculate the test statistics for large datasets.
>
>
> 1.
> Reviewer:
> However, the drawback of the approach is the significantly increased computation due to 1) quadratic complexity for the kernel methods; 2) unable to cache computation among different pairs of nodes (for example, gradient-based approach can compute importance for all node connections in one forward-backward pass). This limits the applicability of the approach to more interesting cases of larger models (for example, models that work on ImageNet) where model compression is of more urgent need. As a result, the significance and impact of the approach is also limited
>
> Response:
> We thank the reviewer for pointing out the computational concerns in larger datasets. Our approach can be implemented in a mini-batch fashion. Basically, we divide data into mini-batches, calculate the test statistics within each mini-batch, and then average the statistics overall mini-batches to produce the test statistics of the whole dataset. The mini-batch size can be smaller than the sub-sampling size and hence reduce the computational cost further. Theoretically, as shown in [1], the type-I error and power are guaranteed when the batch size is greater than the polynomial order of log(n) for the Gaussian kernel.   A similar technique was used in kernel-based deep learning methods, such as GMMN [2] and MMD-GAN [3]. In practice, the performance of the averaged test statistics based on the mini-batches is comparable with the oracle one calculated by using the full data, when the mini-batch size is carefully chosen (see [1] for details). Details of the implementation can be found in our revised manuscript Section 4.4. We report the running time of our method using sub-sampling and mini-batch on a workstation with a 48-core CPU@2.60GHz:
>
> =============================================================
>                                         Lenet-300-100     Lenet-5-Caffe     CIFAR10-VGG16
> sub-sampling (mins)            39.6                     26.8                    312.9
> Mini-batch  (mins)                29.6                     21.8                    103.9
> =============================================================
>
> The computational time of our mini-batch implementation improves dramatically over the sub-sampling implementation, especially for larger datasets such as CIFAR10. Due to the limitation of computational resources, we are not able to complete the experiments on ImageNet. So far the intermediate results look very promising and the final results will be included in our revision. We claim the computation cost is manageable for larger datasets and is of less concern for the following reasons:
> a. Users of model compression techniques (such as ours) only need to compress a model once, which can be conducted on large computer clusters to deal with the computational cost. Then the compressed model can be deployed to many devices with limited memory benefits.
> b.  Our sub-sampling or mini-batch implementations can significantly reduce computational cost.
> c. The statistical testing procedure can be implemented parallelly based on GPUs to further improve computing efficiency.
>
>
> [1] Liu et al. How Many Machines Can We Use in Parallel Computing for Kernel Ridge Regression?, arXiv,  2019.
> [2] Li et al. Generative moment matching networks. NeurlPS. 2015.
> [3] Li et al. "MMD GAN: Towards deeper understanding of moment matching network." NeurlPS. 2017.

---

> ### Author Response · Authors · 2019-11-14
> **reply to Review #2 - 2**
>
>
> 2.
> Reviewer:
> The experiment results are interesting in that it shows the proposed approach can achieve better compression rates given the same test error tolerance on a few small datasets. However, as mentioned above, these experiments are less convincing than more complicated models on larger datasets, such as ImageNet where model compression has greater impact. In addition, the test errors on smaller datasets are easier to achieve, sometimes tuning the optimization settings such as learning rates can result in significant improvement. Therefore, such results are more like preliminary.
>
> Response:
> We added the comparison results for the CIRAR10-VGG16 setting. We include a comparison with
>
> =========================================
>                                             ER% (*)                  CR
> ————————————————————————
> LCR      PCII                         -                             10x
>              Li 17’ [1]                 -                             2.7x
>              Qianghui 18’ [2]   -                             5.8x
>              Babajide 18’ [3]    -                             4.5x
>              Wining Ticket [4]  -                             5x
> ————————————————————————
> MTE     PCII                         6.01 (+1.54)          3x
>              Li 17’ [1]                 6.60 (+0.15)          2.7x
>              Qianghui 18’ [2]   6.67 (-0.60)           5.8x
>              Babajide 18’ [3]     6.33 (-0.13)          4.5x
>              Wining Ticket [4]   6.10 (+0.30)         2x
>              Louizos17’ [5]        8.60 (-1.05)          14x
> =========================================
>
> * the result in the parenthesis is the relative accuracy change compared to its own baseline, larger is better.
>
> ** We reached out to the author of Louizos 2017 [5] about replicating the results. They told us their experiments usedGaussian process HP optimization and took a few weeks to finish. Our approach only took 312.9 mins in this setting.
>
> Again, due to the limited computational resources, we can obtain now, we are not able to complete the experiments on ImageNet. So far the intermediate results look very promising and the final results will be included in our revision.
>
>
> 3.
> Reviewer:
> The paper is general clear and well written. The experiment section should include more important information such as what kernel bandwidth is used for Gaussian kernel, which greatly affects performance. Also, the main text introducing the proposed statistical test is a bit verbose, and dense in unnecessary notations. I think it can be made more succinct by presenting a high level idea (removing mean, three way correlations) first and then the final results. Some intermediate results can be put into Appendix.
>
> Response:
> We revised the section 4 to simplify the notations. Tuning kernel bandwidth optimally still remains an open problem. In this work, we set as e kernel bandwidth is set at the median distance between 10 percent of randomly selected points.
>
>
> [1] Li, Hao, et al. Pruning filters for efficient convnets. arXiv preprint arXiv:1608.08710 (2016).
> [2] Huang, Qiangui, et al. Learning to prune filters in convolutional neural networks. 2018 IEEE Winter Conference on Applications of Computer Vision (WACV). IEEE, 2018.
> [3] Ayinde, Babajide O., and Jacek M. Zurada. Building efficient convnets using redundant feature pruning. arXiv preprint arXiv:1802.07653. 2018.
> [4] Liu, Zhuang, et al. Rethinking the value of network pruning. arXiv preprint arXiv:1810.05270 2018.
> [5] Louizos et al. Bayesian compression for deep Learning. NeurlPS. 2017.

---

### Official Review · AnonReviewer4 · 2019-11-01
**Official Blind Review #4**

**Rating:** 6

**Review:**

This paper introduces a nonparametric score test to estimate the importance of network connections on the final output of Deep Neural Networks. They derive this by modeling the log-transformed joint density of each connection and final output in a tensor product reproducing kernel Hilbert space. They finally derive an asymptotic distribution of the proposed test statistics which only depends on the eigenvalues of the kernel. This importance test is applied to ranking the importance of each connection in Multilayer Perceptron Networks, and Convolutional Networks and sparsifying the networks by removing the least important connections. The method is applied post-training to fully trained networks but evaluated by retraining the sparsified network from scratch. The method is demonstrated in experiments on compressing three networks.

I believe this paper is a borderline accept. It provides a more statistically principled method to examine the importance of connections in a Neural Network and rank them compared to existing compression methods. Due to this ranking, there is a clear method to easily adjust the target compression rate. They are able to achieve high lossless compression rates. However, the benefits shown in the empirical results could be more convincing. It lacks baselines on more complex networks and could benefit from more empirical analysis of the theoretical benefits and properties of this approach.

Pros:
The method is able to maintain accuracy while achieving high compression rates on MNIST and CIFAR10. It does better than the baselines compared to the small networks. They show a capability for increasing generalizability by decreasing error rates.

It provides a well derived statistically principled method to examine the importance of connections and rank them.

Indicates an additional use to visualize the importance of features.

Cons:
The empirical results could be clearer. It lacks baselines for larger models on Cifar10. Could you compare it with the published results of other algorithms? How does it do on larger networks like Imagenet?
Computational efficiency is mentioned but could be examined in greater detail.  How long does it take to run and how is that affected by model size?

The experimental setting is somewhat unclear. The baseline Louizos et al. (2017)  was designed to optimize group sparsity/speed, but the experimental results here only examine the compression rate. Was the baseline run to optimize speed or sparsity?

It would be interesting to examine the correlation of importance score with actual impact on network performance. This might be done with a comparison with random pruning or pruning higher importance connections. It might be useful to examine the performance of the networks after pruning nodes of differing importance without full retraining or just fine-tuning. It is unclear how important full retraining is in this method.

It would be interesting to visualize the importance of features at different depths in the deep convolutional networks.

Minor suggestions: In the introduction, you mention l0 and l1 norm methods, but cite Han et al. (2015) which compares l1 and l2 norm and found l2 norm to be better overall.

typos:
In Abstract: nonparemtric scoring test  -> nonparametric scoring test
In 4.4: sample averarge -> sample average
In 4.5: ASYMPTOTICALLY DISTRIBUTION  ->   ASYMPTOTIC DISTRIBUTION

**Experience Assessment:**

I have read many papers in this area.

**Review Assessment: Checking Correctness Of Derivations And Theory:**

I assessed the sensibility of the derivations and theory.

**Review Assessment: Checking Correctness Of Experiments:**

I carefully checked the experiments.

**Review Assessment: Thoroughness In Paper Reading:**

I read the paper thoroughly.

---

> ### Author Response · Authors · 2019-11-13
> **reply to Review #4 - 1**
>
> We appreciate the reviewer for suggestions and questions. We address the concerns below and updated the manuscript in which we added a mini-batch implementation to efficiently calculate the test statistics for large datasets.
>
> 1.
> Reviewer:
> The empirical results could be clearer. It lacks baselines for larger models on Cifar10. Could you compare it with the published results of other algorithms? How does it do on larger networks like Imagenet? Computational efficiency is mentioned but could be examined in greater detail.  How long does it take to run and how is that affected by model size?
>
> Response:
> We added the comparison results for the CIRAR10-VGG16 setting as follows:
>
>                                             ER% (*)               CR
> ———————————————————————
> LCR      PCII                         -                          10x
>              Li 17’ [1]                 -                          2.7x
>              Qianghui 18’ [2]   -                          5.8x
>              Babajide 18’ [3]    -                          4.5x
>              Wining Ticket [4]  -                          5x
> ———————————————————————
> MTE     PCII                         6.01 (+1.54)       3x
>              Li 17’ [1]                 6.60 (+0.15)       2.7x
>              Qianghui 18’ [2]   6.67 (-0.60)        5.8x
>              Babajide 18’ [3]     6.33 (-0.13)       4.5x
>              Wining Ticket [4]   6.10 (+0.30)      2x
>              Louizos17’ [5]        8.60 (-1.05)       14x
>
> * the result in the parenthesis is the relative accuracy change compared to its own baseline, larger is better.
>
> The computational cost is determined by the number of hypotheses and the sample size. The number of hypothesis tests is equal to the number of edges (in fully connected networks) or the number of filters (CNNs). For each test, the computational time is determined by the sample size. Our approach can be implemented in a mini-batch fashion. Basically, we divide data into mini-batches, calculate the test statistics within each mini-batch, and then average the statistics overall mini-batches to produce the test statistics of the whole dataset. The mini-batch size can be smaller than the sub-sampling size and hence reduce the computational cost further. Theoretically, as shown in [6], the type-I error and power are guaranteed when the batch size is greater than the log(n) order for the Gaussian kernel.   A similar technique was used in kernel-based deep learning methods, such as GMMN [7] and MMD-GAN [8]. In practice, the performance of the averaged test statistics based on the mini-batches is comparable with the oracle one calculated by using the full data, when the mini-batch size is carefully chosen (see [6] for details). We report the running time of our method using sub-sampling and mini-batch on a workstation with a 48-core CPU@2.60GHz:
>
>                                         Lenet-300-100     Lenet-5-Caffe     CIFAR10-VGG16
> sub-sampling (mins)            39.6                     26.8                    312.9
> Mini-batch  (mins)                29.6                     21.8                    103.9
>
> The computational time of our mini-batch implementation improves dramatically over the sub-sampling implementation, especially for larger datasets such as CIFAR10. Due to the limitation of computational resources, we are not able to complete the experiments on ImageNet. So far the intermediate results look very promising and the final results will be included in our revision. We claim the computation cost is manageable for larger datasets and is of less concern for the following reasons:
> a. Users of model compression techniques (such as ours) only need to compress a model once, which can be conducted on large computer clusters to deal with the computational cost. Then the compressed model can be deployed to many devices with limited memory benefits.
> b.  Our sub-sampling or mini-batch implementations can significantly reduce computational cost.
> c. The statistical testing procedure can be implemented parallelly based on GPUs to further improve computing efficiency.
>
>
> [1] Li, Hao, et al. Pruning filters for efficient convnets. arXiv preprint arXiv:1608.08710 (2016).
> [2] Huang, Qiangui, et al. Learning to prune filters in convolutional neural networks. 2018 IEEE Winter Conference on Applications of Computer Vision (WACV). IEEE, 2018.
> [3] Ayinde, Babajide O., and Jacek M. Zurada. Building efficient convnets using redundant feature pruning. arXiv preprint arXiv:1802.07653. 2018.
> [4] Liu, Zhuang, et al. Rethinking the value of network pruning. arXiv preprint arXiv:1810.05270 2018.
> [5] Louizos et al. Bayesian compression for deep Learning. NeurlPS. 2017.
> [6] Liu et al. How Many Machines Can We Use in Parallel Computing for Kernel Ridge Regression?, arXiv,  2019.
> [7] Li et al. Generative moment matching networks. NeurlPS. 2015.
> [8] Li et al. MMD GAN: Towards deeper understanding of moment matching network. NeurlPS. 2017.

---

> ### Author Response · Authors · 2019-11-14
> **reply to Review #4 - 2**
>
>
> 2.
> Reviewer:
> The experimental setting is somewhat unclear. The baseline Louizos et al. (2017)  was designed to optimize group sparsity/speed, but the experimental results here only examine the compression rate. Was the baseline run to optimize speed or sparsity?
>
> Response:
> We reached out to the author of Louizos 2017 [1] about replicating the results, they told us their experiments are achieved by Gaussian process HP optimization running for a few weeks.
>
>
> 3.
> Reviewer:
> It would be interesting to examine the correlation of importance score with actual impact on network performance. This might be done with a comparison with random pruning or pruning higher importance connections. It might be useful to examine the performance of the networks after pruning nodes of differing importance without full retraining or just fine-tuning. It is unclear how important full retraining is in this method.
>
> Response:
> We thank the reviewer for the suggestions. We compare the performance of pruning with fine-tuning, random pruning and our proposed approach with respect to the LCR as follows:
>
> =====================================================
>                                          Lenet-300-100          Lenet-5-Caffe
> Random pruning          1x                                1x
> Fine-tuning                    11x                              17x
> Retraining-pruning       26x                             38x
> =====================================================
>
> The results show that random pruning can not achieve lossless compression. Fine-tuning has a lower LCR compared with our proposed retraining-pruning method. An intuitive explanation is that, in our proposed method, the retraining step keeps the same initial weights for the pruned network which gives more freedom for the algorithm to adapt to higher accuracy.
>
>
> 4.
> Reviewer:
> It would be interesting to visualize the importance of features at different depths in the deep convolutional networks.
>
> Response:
> We have a visualization of the connections between the input and the first hidden layer to interpretation the contribution of input features in our manuscript; however, due to the unidentifiable issue of the hidden layers, the connections between the hidden layers at different depths are hard to interpret.
>
>
> 5.
> Reviewer: Minor suggestions: In the introduction, you mention l0 and l1 norm methods, but cite Han et al. (2015) which compares l1 and l2 norm and found l2 norm to be better overall.
>
> Response:
> We thank the reviewer for pointing out the l2 norm approach. We revised our paper to add reviews of the l2 norm-based approach.
>
>
> [1] Louizos et al. Bayesian compression for deep Learning. NeurlPS. 2017.

---

> > ### Comment · AnonReviewer4 · 2019-11-14
> > **Questions about the random pruning experiments**
> >
> > I had some clarifying questions about the random pruning experiments. Were the fine-tuning results with the random pruning? For a fair comparison, the treatment of the pruned network whether using random pruning or your importance metric should be the same. Did you try pruning with your method without retraining?
> >
> > It seems like the most drastic results should be shown with fine-tuning of an importance pruned and a randomly pruned network

---

> > > ### Author Response · Authors · 2019-11-14
> > > **reply to Reviewer 4**
> > >
> > > Thank you for the clarification. In our retraining step, as shown in step 6 in Algoirthm 1, we set the same initial value for non-zero weights (or filters) and retrain the sparsified DNN. This is different from assigning a new random initial value for non-zero weights. In the updated comparison, we use our proposed importance metric for the fine-tuning. It shows that our retraining set-up has better performance than fine-tuning.

---

> > > > ### Comment · AnonReviewer4 · 2019-11-15
> > > > **Random pruning questsions**
> > > >
> > > > I was wondering about the Random pruning results you showed above in particular. Was '``Fine-tuning' using random pruning or importance metric. When you found that random pruning couldn't achieve LCR of less than one was that with or without fine tuning? The treatment of the pruned network whether using random pruning or your importance metric should be the same.

---

> > > > > ### Author Response · Authors · 2019-11-15
> > > > > **Reply to reviewer 4**
> > > > >
> > > > > In the above experiment, the random pruning is a naive approach by randomly setting weights to zero and than retrain the network with same initials for nonzero weight. Thus, it has same treatment with our proposed method. The only difference is that we use the proposed importance metric and random pruning does not use any importance metric.

---

### Decision · Program_Chairs · 2019-12-19

**Decision:**

Accept (Poster)

**Comment:**

This paper proposes a novel approach for pruning deep neural networks using non-parametric statistical tests to detect 3-way interactions among two nodes and the output. While the reviewers agree that this is a neat idea, the paper has been limited in terms of experimental validation. The authors provided further experimental results during the discussion period and the reviewers agree that the paper is now acceptable for publication at ICLR-2020.